# Enrichment of Trypsin Inhibitor from Soybean Whey Wastewater Using Different Precipitating Agents and Analysis of Their Properties

**DOI:** 10.3390/molecules29112613

**Published:** 2024-06-02

**Authors:** Yongsheng Zhou, Siyun Zhou, Cuiwen Lu, Yihao Zhang, Haiyan Zhao

**Affiliations:** College of Food and Biochemical Engineering, Guangxi Science & Technology Normal University, Laibin 546199, China

**Keywords:** trypsin inhibitor, soybean whey, ethanol precipitation

## Abstract

Recovering valuable active substances from the by-products of agricultural processing is a crucial concern for scientific researchers. This paper focuses on the enrichment of soybean trypsin inhibitor (STI) from soybean whey wastewater using either ammonium sulfate salting or ethanol precipitation, and discusses their physicochemical properties. The results show that at a 60% ethanol content, the yield of STI was 3.983 mg/mL, whereas the yield was 3.833 mg/mL at 60% ammonium sulfate saturation. The inhibitory activity of STI obtained by ammonium sulfate salting out (A-STI) was higher than that obtained by ethanol precipitation (E-STI). A-STI exhibited better solubility than E-STI at specific temperatures and pH levels, as confirmed by turbidity and surface hydrophobicity measurements. Thermal characterization revealed that both A-STI and E-STI showed thermal transition temperatures above 90 °C. Scanning electron microscopy demonstrated that A-STI had a smooth surface with fewer pores, while E-STI had a rough surface with more pores. In conclusion, there was no significant difference in the yield of A-STI and E-STI (*p* < 0.05); however, the physicochemical properties of A-STI were superior to those of E-STI, making it more suitable for further processing and utilization. This study provides a theoretical reference for the enrichment of STI from soybean whey wastewater.

## 1. Introduction

Soybean, renowned for its rich oil and protein content, is a significant source of protein and fat nutrients for humans [1]. It is commonly called the “king of beans” due to its abundant nutrients and high nutritional value, making it highly sought after worldwide [1]. In 2020, the worldwide market value of soybean reached USD 127.81 billion and is expected to reach approximately USD 162 billion by 2027 [2]. Soybean protein isolate (SPI) is a deep-processing product derived from defatted soybean meal. Due to its excellent nutritional profile, functional properties, and health benefits, SPI exhibits significant potential as a valuable food ingredient [3]. SPI is typically obtained through alkali extraction-acid precipitation, involving the extraction of defatted soybean meal under alkaline conditions (pH 7.5~pH 8.5), followed by precipitation under acidic conditions (pH 4.5) [3]. The production of SPI results in the generation of significant quantities of soybean whey by-products [4]. On average, 20 tons of soybean whey are generated per ton of SPI produced [4]. Soybean whey contains various nutrients, including monosaccharides, oligosaccharides, and soybean whey protein [5]. As soybean whey is rich in sugars, proteins, minerals, and various nutrients, it is prone to microbial growth and can develop an unpleasant odor when stored under natural conditions [6]. Consequently, the timely disposal or management of soybean whey becomes necessary. The direct discharge or traditional biochemical treatment of soybean whey can adversely affect the environment and waste valuable resources, which is incompatible with sustainability strategies [6,7]. Directly discharging soybean whey into water can lead to eutrophication. This nutrient-rich wastewater promotes the rapid growth of algae and bacteria, which, in turn, depletes the oxygen levels in the water. Consequently, this can create anaerobic conditions, harming aquatic animals and plants and potentially causing large-scale death. This process ultimately degrades the water quality of the river and adversely affects the surrounding ecological environment [6]. Consequently, the adoption of an appropriate strategy for the integrated utilization of soybean whey becomes of paramount importance. By doing so, we can contribute to environmental preservation, resource conservation, and cost reduction for companies. Tu et al. conducted a study utilizing water kefir grains for fermenting soybean whey to transform it into novel functional food ingredients through microbial fermentation or enzyme-catalyzed reactions [8]. The findings demonstrated a significant increase in total flavonoids, total phenols, and isoflavone glycosides in soybean whey following fermentation. Moreover, fermentation resulted in the generation of numerous new aromatic volatile compounds, thereby enhancing the sensory attributes of soybean whey [8]. Dai et al. also used *Cordyceps militaris* SN-18 to ferment soybean whey. The fermentation increased the content of essential amino acids, total phenolics, flavonoids, and isoflavone glycosides while reducing oligosaccharides [9]. In addition, the recovery of valuable compounds from soybean whey, such as soybean oligosaccharides, soybean whey protein, soybean isoflavones, and polyphenols, represents an advantageous approach for energy conservation and emission reduction. However, significant challenges persist due to the high cost and inadequate motivation to recover organic substances.

Soybean whey protein is an acid-soluble protein, mainly composed of 2S protein and 7S protein, accounting for approximately 9.0% to 15.3% of the total soybean seed protein. It contains various components such as soybean trypsin inhibitor (STI), including Kunitz trypsin inhibitor (KTI) and Bowman–Birk trypsin inhibitor (BBI), soybean lectin, lipid oxidase, β-amylase, and other fractions [10]. Enriching STI from soybean whey wastewater can reduce organic matter and minimize environmental pollution. Moreover, STI exhibits specific physiological functions, such as anti-cancer, anti-inflammatory, and anti-bacterial activities, thereby possessing significant development potential in the health food and biopharmaceutical markets [11,12,13].

Currently, the main methods for crude protein separation are the isoelectric point method, salting out, and organic solvent precipitation [14]. The isoelectric point precipitation method relies on the principle that when the pH of an amphoteric electrolyte reaches its isoelectric point, the electrostatic charge on the molecular surface becomes zero. This leads to the weakening or destruction of the stable double-electric layer and hydration film, resulting in increased intermolecular attraction and decreased solubility. As a result, proteins aggregate and precipitate out [15]. Soybean whey wastewater is a by-product obtained from the precipitation of SPI using the isoelectric point method [3]. Moreover, the isoelectric point of STI closely resembles that of SPI. Consequently, the conventional isoelectric point precipitation method proves ineffective for the recovery of STI from soybean whey wastewater. An alternative approach known as the “salting out method” involves the introduction of inorganic salts into the protein solution. This addition leads to the mutual neutralization of salt ions and protein molecules’ surface charges, reducing the electrostatic interactions between the proteins. On the other hand, salt ions mediate the hydrophobic interaction of the proteins, reducing their solubility and leading to the aggregation and precipitation of proteins. The organic solvent precipitation method uses organic solvents to reduce the dielectric constant of aqueous solutions, thereby increasing the attraction between two oppositely charged groups. This process destroys the hydrophobic membrane of proteins, reducing their hydrophilicity and resulting in the aggregation and precipitation of protein molecules [16]. Organic solvent precipitation is a commonly used and scalable method for protein separation. The discriminatory capacity of this method is higher than that of salinization, as precipitation does not require desalination, filtration is easier, the conditions are mild and inexpensive, and the process is simple. Nevertheless, it is essential to acknowledge that protein solutions treated with organic solvents may cause the irreversible denaturation of proteins and the loss of their original functional properties. Consequently, carefully selecting an appropriate precipitating agent to recover STI from soybean whey is potentially instructive.

In this paper, soybean trypsin inhibitor (STI) was enriched from soybean whey wastewater using two different methods: ammonium sulfate salting out and ethanol precipitation. We compared these enrichment methods to determine the most effective extraction technique for STI from soybean whey wastewater, providing a scientific basis for selecting the optimal method. Subsequently, we examined the physicochemical properties of the obtained STI, including solubility, thermal properties, and surface morphology, which are critical for its application and further processing. The results of this study offer a valuable theoretical reference for the efficient enrichment of STI from soybean whey wastewater.

## 2. Results and Discussion

### 2.1. Analysis of Yield and Protein Recovery Rate

Soybean whey powder was obtained after soybean whey freeze-drying (Table 1). In total, 24.90 mg of soybean whey powder was obtained per 100 mL of soybean whey. The analysis of the freeze-dried samples showed that the protein content accounted for 17.19% of the total solids.

Inorganic salt ammonium sulfate and organic solvent ethanol were used to enrich STI from soybean whey. The results, as presented in Table 1, demonstrated a gradual increase in the yields of STI with an increase in ammonium sulfate saturation or ethanol percentage. In all cases, the obtained STI exhibited a high protein content, exceeding 90%. The yield of STI was found to be 0.317 mg/mL with a minimum protein recovery of 7.06% at 20% ammonium sulfate saturation and 3.983 mg/mL with a maximum protein recovery of 87.02% at 60% ethanol content. Notably, at 60% ammonium sulfate saturation, the yield of STI was 3.833 mg/mL, and the protein recovery was 83.39%. At this point, no significant difference was observed in the yield of STI enriched using the ethanol precipitation method compared to the ammonium sulfate salting out method.

### 2.2. STI Composition Analysis

STI was analyzed by SDS-PAGE, and the results are shown in Figure 1. The SDS-PAGE analysis revealed that the STI was mainly composed of proteins with molecular weights smaller than 35 kDa. As shown in Figure 1a, when the ammonium sulfate saturation was below 40%, the grayness of the bands corresponding to a molecular weight less than 20 kDa gradually deepened as the ammonium sulfate saturation increased, indicating that the purity of the STI increased [17]. However, when the ammonium sulfate saturation was higher than 40%, with the increase in the ammonium sulfate saturation, several bands with a molecular weight greater than 25 kDa appeared, while the protein bands with a molecular weight less than 20 kDa did not significantly deepen in grayness, indicating that exceeding 40% ammonium sulfate saturation might lead to a decrease in the STI purity. Furthermore, Figure 1b displays the effect of increasing ethanol percentage on STI composition. As the ethanol percentage increased, the grayness of protein bands with molecular weights less than 20 kDa gradually decreased. Conversely, bands with molecular weights greater than 25 kDa exhibited increased intensity and deep grayness. These observations indicated that an elevated ethanol percentage might lead to a reduction in STI purity.

### 2.3. STI Purity Analysis

Gel filtration chromatography is a separation technique based on the size and shape of proteins, often used for further purification of crudely isolated proteins [18]. Figure 2 shows the chromatograms of the samples dissolved in 0.2 M phosphate-buffered solution, passing through a Superdex 75 Increase 10/300 gel filtration pre-load column. Figure 2a shows the gel filtration chromatograms of KTI (sigma, T9128), BBI (sigma, T9777), and A-STI. The results indicated that the peak of KTI (20 kDa) appeared at 22 min (11 mL mobile phase elapsed) and diminished after 26 min (13 mL mobile phase), while the peak of BBI (8 kDa) appeared at 28 min (14 mL mobile phase) and disappeared after 30 min (15 mL mobile phase). Therefore, based on the retention time of the KTI and BBI peaks, the purity of the STI obtained with different ammonium sulfate saturation levels can be tentatively determined. As shown in Table 2, the purity of A-STI was above 68%. When the ammonium sulfate saturation was 40%, there was 81.77% STI and 18.23% hetero protein, with the highest purity of STI found in the concentrate; this result is consistent with the SDS-PAGE results. Figure 2b shows the gel filtration chromatograms of KTI, BBI, and E-STI. The results demonstrate that the total peak area of STI gradually decreased with the increase in the ethanol percentage, indicating that less STI was soluble in the 0.2 M phosphate-buffered solution. This observation may be related to the structural changes in the STI caused by ethanol, impacting the solubility, and the high ethanol concentration leading to protein denaturation [19]. A high concentration of ethanol solution can induce the exposure of hydrophobic groups, thus decreasing the solubility of proteins. Moreover, we observed that the total peak area of the E-STI chromatogram was generally smaller than that of A-STI, indicating that the solubility of the protein precipitated by ethanol precipitation may be inferior to that of the protein obtained by ammonium sulfate precipitation. Table 2 indicates that the purity of E-STI was consistently above 60%. When the ethanol percentage was 60%, the highest purity of STI was 78.79%. However, this contrasts with the results of SDS-PAGE, possibly because, after treatment with a high ethanol percentage, larger protein molecules (>20 kDa) aggregated together, leading to their exclusion from the mobile phase after passing through the 0.22 μm filter membrane. This phenomenon contributes to the high purity of STI, corresponding to the small total peak area of E-STI.

### 2.4. Analysis of STI Activity

As shown in Figure 3, the trypsin-inhibitory activity of A-STI was found to be stronger than that of E-STI. Figure 3a illustrates that the trypsin inhibitory activity of STI initially increased and then decreased with increasing ammonium sulfate saturation. A-STI exhibited a maximum trypsin inhibitory activity of 1129 U/mg at 40% ammonium sulfate saturation, possibly attributed to its highest purity level. This result aligns with the findings of the electrophoresis and gel filtration chromatography. Figure 3b shows that the inhibitory activity of STI showed a fluctuating trend (decreasing, then increasing, and then decreasing) as the percentage of ethanol increased. This result may be related to ethanol changing the protein structure. It was shown that the increase in ethanol content would lead to irreversible changes in protein spatial structure, consequently impacting their physicochemical properties and diminishing or even eliminating their original physiological and functional attributes [19].

### 2.5. Effect of Temperature and pH on the Solubility of STI

Solubility is a physical property of substances forming solutions and refers to the ability of a substance to dissolve in a particular solvent. Proteins, as organic macromolecular compounds, exist in a dispersed state (colloidal state) in water. Therefore, the concept of the solubility of proteins in water becomes more nuanced, referring to the degree or extent of protein dispersion within the aqueous medium [20]. The protein solubility characteristics are significant in practical applications, particularly in the context of the extraction, separation, and purification of natural proteins. Changes in protein solubility behavior can also serve as indicators of protein denaturation, in addition to the fact that the application of proteins in beverages is directly related to their solubility properties. As a direct consequence, understanding the solubility of STI holds crucial importance in their isolation and purification and in comprehending their structural and functional properties [21,22].

Figure 4 shows the solubility curves of A-STI and E-STI under different pH and temperature conditions. In Figure 4a, the solubility of A-STI exhibited a characteristic trend of first decreasing and then increasing with the increase in pH within the pH range of 3.0 to 10.0. Notably, the minimum solubility of A-STI occurred at pH 5.0, measuring 67.70%, while the maximum solubility was achieved at pH 10.0, reaching 100%. Similarly, the solubility curves of E-STI showed the same trend as that of A-STI, with decreasing and then increasing patterns. The solubility of E-STI reached its minimum at pH 4.0, measuring 14.59%, and reached its peak at pH 10.0, attaining 75.77%. In summary, the results indicated that the solubility of A-STI was higher than that of E-STI at the same pH value within the pH range of 3.0 to 10.0. This disparity in solubility could be attributed to protein denaturation induced by the high ethanol content. Moreover, A-STI exhibited its lowest solubility at pH 5.0, while E-STI exhibited the lowest solubility at pH 4.0, a phenomenon potentially related to the isoelectric point of STI, which typically corresponds to a protein’s lowest solubility state [20].

Protein activity is generally weak at low temperatures; the lower the temperature, the weaker the activity. Increasing the temperature within an appropriate range can significantly enhance protein activity [23]. However, it is essential to note that higher temperatures may induce changes in protein structure, potentially affecting the physicochemical properties of the protein. Figure 4b shows the effect of temperature on the solubility of STI. The results indicated that the solubility of A-STI decreased with the increase in temperature (25 °C→45 °C) and then stabilized (45 °C→85 °C), followed by a subsequent decrease (85 °C→95 °C). Throughout the process, a slight overall reduction in solubility was observed with increasing temperature. In contrast, the solubility of E-STI exhibited an initial increase (25 °C→35 °C) followed by stabilization (35 °C→95 °C). Although the effect of temperature on the solubility of both A-STI and E-STI is not particularly significant, it is evident that temperature does influence their solubility characteristics.

### 2.6. Effect of Temperature and pH on the Turbidity of STI

Turbidity indicates the degree of obstruction caused by suspended particles to light transmission in water [24]. When proteins are dispersed in water, their particle size, shape, and surface area affect light transmission. Therefore, the turbidity value can visually reflect the state of the protein in water. Figure 5 shows the turbidity curves of A-STI and E-STI under various pH and temperature conditions. Figure 5a shows that the absorbance values of both A-STI and E-STI initially increased and then decreased within the pH range of 3.0 to 10.0, reaching their maximum at pH 5.0. This phenomenon indicates that STI at pH 5.0 exhibited excellent particle density, leading to increased light obstruction and, consequently, reflecting its lower solubility, which aligns with the results in Section 3.5. Figure 5b shows that the absorbance value of A-STI gradually increased with the temperature increase from 25 °C to 95 °C, indicating that temperature affects the particle size of STI. Higher temperatures result in reduced light transmission through the protein solution, indicating a decline in STI solubility, which is consistent with the findings presented in Section 3.5. On the other hand, the absorbance value of E-STI showed a trend of increasing and then decreasing with the increase in temperature, indicating that the appropriate increase in temperature can lead to enhanced particle size reduction in E-STI and improved solubility. However, these results are inconsistent with the results in Section 3.5, possibly because the increase in temperature accelerates the aggregated precipitation of particles, altering the absorbance value of STI. When comparing the turbidity of A-STI and A-STI, we find that the absorbance value of A-STI is larger than that of E-STI. This observation may be attributed to A-STI’s smaller but more numerous particles, whereas E-STI exhibits larger particles and reduced dispersion, rendering it more prone to precipitation.

### 2.7. Surface Hydrophobicity Analysis of STI

When the conditions of excitation intensity, wavelength, solvent, and temperature are fixed, the intensity of the emitted light of a substance is proportional to its concentration in solution within a specific concentration range, making it suitable for quantitative analysis. Hydrophobicity is the repulsive force between a non-polar substance and a polar environment. Thermodynamically, it represents the high or low energy required to dissolve a non-polar substance in water or the tendency of the substance to self-aggregate in the aqueous phase, and the greater the tendency for self-aggregation, the stronger the hydrophobicity [25].

Surface hydrophobicity is an essential property of proteins, indicating the physical property of mutual repulsion between protein molecules and water. This characteristic is pivotal in maintaining protein stability, influencing their conformation, and determining their functional activity [26]. Moreover, protein solubility is linked to its surface hydrophobicity, wherein increased hydrophobicity corresponds to reduced protein solubility and vice versa [25]. Therefore, determining the surface hydrophobicity of proteins helps in understanding the interactions between protein molecules, between proteins and other non-protein molecules, and in comprehending the changes in protein properties in different solution environments. Figure 6a,b show the fluorescence intensity plots of A-STI and E-STI at different concentrations, respectively. The results demonstrated that the emission light intensity of both A-STI and E-STI gradually decreased with the decrease in the STI concentration, exhibiting a proportional relationship. Notably, at the same concentration of STI, the emitted light intensity of A-STI was greater than that of E-STI. Figure 6c,d show the slope of the fluorescence intensity as a function of protein concentration for A-STI and E-STI, respectively. The results showed that the surface hydrophobicity index S1 of the A-STI was 528,143.1, while the surface hydrophobicity index S2 of the E-STI was 6,680,619.2. The higher value of S2 compared to S1 indicates that E-STI possesses a more elevated surface hydrophobicity than A-STI. This corresponds to the solubility of A-STI compared with that precipitated by ethanol precipitation. This observation aligns with the solubility characteristics of A-STI compared to that of E-STI, particularly in the context of ethanol precipitation.

### 2.8. Thermal Characterization of STI

Differential Scanning Calorimetry is a widely used technique for assessing the heat absorption and exothermic processes of various materials. In the food field, DSC finds application in characterizing the thermal denaturation temperature of proteins. Figure 7 shows the thermal stability of STI. The results revealed that the thermal stability patterns of A-STI and E-STI exhibited negative single peaks, with both variants demonstrating similar behavior. The thermal transition temperature observed for A-STI was 92.01 °C, while E-STI displayed a slightly higher thermal transition temperature of 93.38 °C. These findings collectively indicate that both A-STI and E-STI possess elevated thermal stability. Moreover, it is noteworthy that the thermal stability of E-STI appears slightly superior to that of A-STI.

### 2.9. Surface Structure Analysis of STI

Figure 8 presents the schematic scanning electron microscopy images of STI obtained from soybean whey wastewater using different precipitating agents. Among these images, Figure 8a-1 (A-STI) and Figure 8b-1 (E-STI) show that the STI obtained by different precipitating agents were observed in general agreement at a field of view of 200× magnification, and the microstructure of STI appeared fragmented. However, at a higher magnification of 20,000×, discrepancies in the observed results become apparent. The A-STI (Figure 8a-2) exhibited a smooth surface structure with relatively fewer pores, while the E-STI (Figure 8b-2) displayed a rough surface structure and more pores.

## 3. Materials and Methods

### 3.1. Feedstocks and Reagents

The Yuwang Ecological Food Industry Co., Ltd. (Dezhou, China) provided a cooled, defatted soybean meal stored at 0~4 °C. The protein marker and loading buffer were obtained from Beijing Solarbio Technology Co., Ltd. (Beijing, China). The trypsin (BAEE > 10,000 Units/mg) was purchased from Shanghai Macklin Biochemical Co., Ltd. (Shanghai, China). The benzoyl-l-arginine-p-nitroaniline (BAPA), KTI (T9128), and BBI (T9777) were obtained from Sigma Chemical Co., Ltd. (St. Louis, MO, USA). All other reagents were of analytical grade.

### 3.2. Preparation of the Soybean Whey

Soybean whey was prepared according to previous procedures with some modifications [3]. The detailed steps were as follows: Mixing the defatted soybean meal with deionized water at a ratio of 1:10 (*w*/*v*), using 1 M NaOH to adjust the pH of the mixture to 7.5, extracting in a constant temperature water bath at 50 °C for 50 min, and centrifuging at 4000 rpm for 20 min to achieve solid/liquid separation. The supernatant was adjusted to pH 4.5 using 20% HCl and centrifuged to obtain the soybean whey.

### 3.3. Preparation of Soybean Whey Concentrate

The soybean whey obtained from Section 2.2 was added into a rotary evaporation flask and evaporated at 40 °C. Part of the water was removed to obtain the soybean whey concentrate. The obtained soybean whey concentrate volume was 1/3 of the original soybean whey.

### 3.4. Enrichment of STI by Ammonium Sulfate Precipitation

Ammonium sulfate was slowly added to the soybean whey concentrate to achieve different saturation levels: 20%, 30%, 40%, 50%, and 60%, respectively. After standing for 120 min at 4 °C, the solid–liquid separation was performed by centrifuging the mixture at 4000 rpm for 20 min. The resulting precipitate was dissolved using deionized water. The solutions were then dialyzed overnight with dialysis bags with a molecular weight cutoff (MwCO) of 1000 Da (MYM biological technology company limited, Chicago, IL, USA). During dialysis, the deionized water was replaced 2–3 times. The dialysate was freeze-dried and weighed, and the precipitates obtained from each saturation level were marked as A-20%, A-30%, A-40%, A-50%, and A-60%, respectively [3].

### 3.5. Enrichment of STI by Ethanol Precipitation Method

Anhydrous ethanol was slowly added to the soybean whey concentrates to achieve different ethanol content percentages: 20%, 30%, 40%, 50%, and 60% (*v*/*v*), respectively. After standing for 120 min at 4 °C, the solid–liquid separation was performed by centrifugation at 4000 rpm for 20 min. The resulting precipitate was dissolved using deionized water. The solutions were then dialyzed overnight using dialysis bags. During dialysis, the deionized water was replaced 2–3 times. The dialysate was freeze-dried and weighed, and the precipitates obtained from each ethanol content level were marked as E-20%, E-30%, E-40%, E-50%, and E-60%, respectively [27,28].

### 3.6. Analytical Method

#### 3.6.1. Calculation of STI Yield

$$Yield\;(\mathrm{mg}/\mathrm{mL})=\frac{m}{v}\times 100\%$$
where m: the mass (mg) of the soybean trypsin inhibitors; v: the volume (mL) of the soybean whey.

#### 3.6.2. Determination of STI Protein Content

The total crude protein of STI was evaluated using automatic Kjeldahl analysis equipment (Kjeltec™ 8400, FOSS, Hillerød, Denmark) and calculated using a protein conversion factor of 5.71 [29].

#### 3.6.3. Analysis of STI Composition by SDS-PAGE

The analysis of STI composition was performed using SDS-PAGE under reducing conditions on a mini-vertical electrophoresis system (Bio-Rad, Hercules, CA, USA) [3,30]. Here, 15% separation gel and 5% concentrated gel were prepared. The different samples were loaded onto the gel at 40 μg. After electrophoresis, a Coomassie brilliant blue G-250 solution was used for staining, and the decolorizing solution was used to remove excess stain. The samples were scanned using a gel electrophoresis imager system (Bio-Rad, Hercules, CA, USA).

#### 3.6.4. Analysis of STI Purity by Gel Filtration Chromatography

STI purity was analyzed using an AKTA protein purification system (Marlborough, MA, USA) [28]. Superdex 75 Increase 10/300 gel filtration pre-loaded columns were equilibrated with three column volumes of 0.2 M phosphate buffer at pH 7.4. STI was re-dispersed in 0.2 M phosphate buffer at a concentration of 2.0 mg/mL. The sample solution was passed over a 0.22 μm aqueous membrane and slowly injected into the AKTA protein purifier with a syringe. The AKTA protein purifier operated at a flow rate of 0.5 mL/min for elution with a detection wavelength of 220 nm.

#### 3.6.5. Determination of the Trypsin Inhibitory Activity

The prepared sample solution (1 mL) and Tris-HCl buffer solution (1 mL) were pipetted into 15 mL tubes. After incubation in a water bath at 37 °C for 10 min, 2.5 mL of a BAPA solution at a concentration of 0.4 mg/mL was added to each tube and mixed, followed by the addition of 1.0 mL of trypsin solution at a concentration of 20 μg/mL. The mixture was allowed to react at 37 °C for exactly 10 min, after which 0.5 mL of acetic acid (30% *v*/*v*) was added to terminate the reaction. A negative control was prepared by adding trypsin only after adding acetic acid [3]. The trypsin inhibitor activity was then calculated by using the following equation:$$Trypsin\;inhibitor\;activity/(TIU)=\left[\left(Ar-Abr\right)-\left(As-Abs\right)\right]\times \frac{F}{0.02}$$
where *Ar* is the absorbance of the standard solution, *Abr* is the absorbance of the standard blank solution, *As* is the absorbance of the sample solution, *Abs* is the absorbance of the sample blank solution, and *F* is the dilution multiple.

#### 3.6.6. Determination of Solubility

STI solubility was measured using a previously described method with slight modifications [30]. STI (100 mg) was dispersed into 10 mL of deionized water. The pH was adjusted using HCl or NaOH, and the sample suspensions were thoroughly mixed using a magnetic stirring device at different temperatures for 2 h. Temperature (25, 35, 45, 55, 65, 75, 85, 95 °C) and pH (3, 4, 5, 6, 7, 8, 9, 10) were considered. The suspension was centrifuged at 4000 rpm for 10 min to separate the insoluble residues. The protein content of the supernatant was determined using the BCA protein assay kit.

#### 3.6.7. Determination of Turbidity

STI (100 mg) was dispersed into 10 mL of deionized water. The pH was adjusted using HCl or NaOH, and the sample suspensions were thoroughly mixed using a magnetic stirring device at different temperatures for 2 h. Temperature (25, 35, 45, 55, 65, 75, 85, 95 °C) and pH (3, 4, 5, 6, 7, 8, 9, 10) were considered. The suspensions were diluted 5-fold with deionized water and measured by ultraviolet spectrophotometer (Agilent, Santa Clara, CA, USA) at OD600 [31].

#### 3.6.8. Determination of Surface Hydrophobicity

STI samples were serially diluted with 20 mM phosphate buffer at pH 7.4 to obtain different concentrations (0.0625, 0.1250, 0.2500, 0.5000, 1.0000 mg/mL). Then, 20 μL of 8-Anilino-1-naphthalenesulfonic acid (ANS) solution (8 mM ANS in 20 mM phosphate buffer) was added to 4 mL of each sample and shaken for 3 min in the dark. The slope of the fluorescence intensity as a function of protein concentration was used as an index indicating the hydrophobicity surface of the protein using an FS5 fluorescence spectrometer (Edinburgh Instruments, Livingston, UK) with a slit of 1 nm, an excitation wavelength of 260 nm, and an emission wavelength range of 280 nm~420 nm [29,32].

#### 3.6.9. Thermal Characteristics of STI by Differential Scanning Calorimetry

STI (5 mg) was placed in a solid sample crucible while a blank crucible was prepared. The blank crucible was placed on the left side, and the sample crucible containing STI was placed on the right side for DSC analysis with an initial temperature of 40 °C, ramped up at 10 °C/min, and ending at 200 °C.

#### 3.6.10. Morphology of STI by Scanning Electron Microscopy (SEM)

The microstructure of STI was observed using the field-emission SEM (Zeiss Merlin Compact, Jena, Germany) [31]. The samples were sprayed with gold for 45 s using an Oxford Quorum SC7620 sputter coater, followed by a Zeiss Merlin Compact scanning electron microscope to photograph the sample morphology with an accelerating voltage of 3 kV during morphology.

### 3.7. Data Analysis

SPSS Statistics 19 software (IBM, Chicago, IL, USA) was used for data processing. The experimental data were expressed as “X ± SD”. Significant differences between measurements were set at *p* < 0.05. Origin 2021 software was used for data visualization.

## 4. Conclusions

In this study, two methods, namely, ammonium sulfate salting out and ethanol precipitation, were successfully employed for enriching STI from soybean whey wastewater. The yield of STI demonstrated a gradual increase with the increase in ammonium sulfate saturation or ethanol content. At 60% ammonium sulfate saturation and 60% ethanol content, the yield of STI surpassed 3.80 mg/mL, with a protein recovery rate exceeding 80%. Regarding purity and trypsin inhibitory activity, the highest levels were achieved at 40% ammonium sulfate saturation. Comparing the solubility of the obtained STI variants, A-STI exhibited better solubility than E-STI. Both A-STI and E-STI exhibited high thermal stability, surpassing 90 °C. Microscopic analysis revealed that both A-STI and E-STI manifested as flakes. However, E-STI showed a rough surface structure with increased pores compared to A-STI.

In future research, we plan to optimize the ammonium sulfate salting out method to enhance the quality and yield of STI from soybean whey wastewater. Additionally, we will explore the potential applications of STI in the food industry.

## Figures and Tables

**Figure 1 molecules-29-02613-f001:**
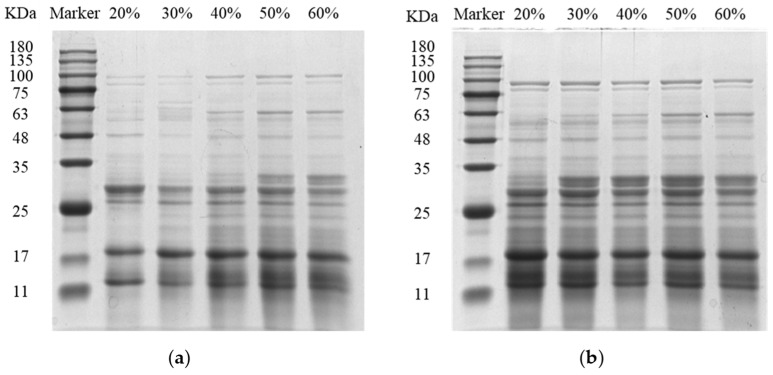
Electrophoretic diagram of precipitated protein with different ammonium sulfate saturation (**a**) and ethanol content (**b**).

**Figure 2 molecules-29-02613-f002:**
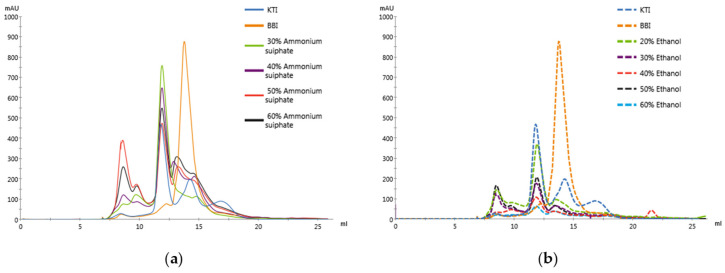
Gel filtration chromatogram of precipitated protein with different ammonium sulfate saturation (**a**) and ethanol content (**b**).

**Figure 3 molecules-29-02613-f003:**
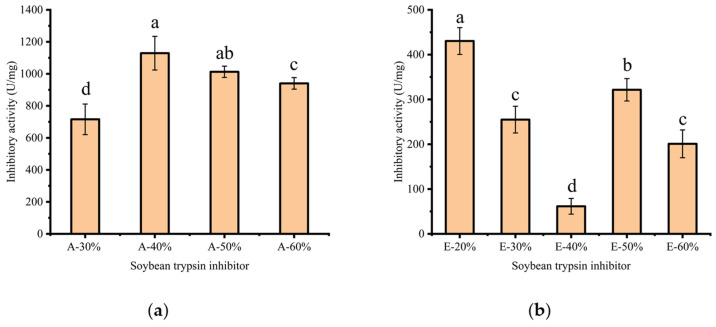
Analysis of STI activity from different methods (A: ammonium sulfate; E: ethanol); Inhibitory activity of STI obtained with different ammonium sulfate saturation (**a**); Inhibitory activity of STI obtained with different ethanol content (**b**); Different lowercase letters indicate significant differences (*p* < 0.05) among the different conditions.

**Figure 4 molecules-29-02613-f004:**
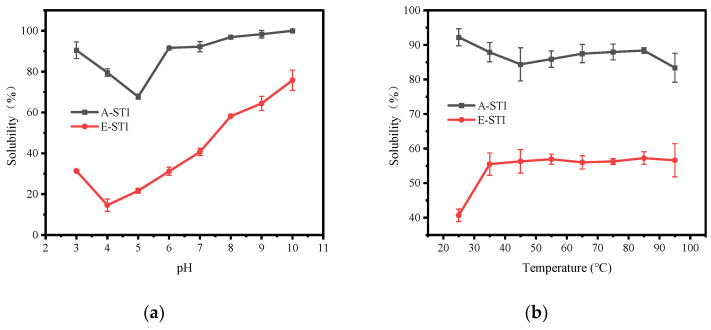
Effect of temperature and pH on the solubility of STI: (**a**) effect of pH on solubility of STI; (**b**) effect of temperature on solubility of STI.

**Figure 5 molecules-29-02613-f005:**
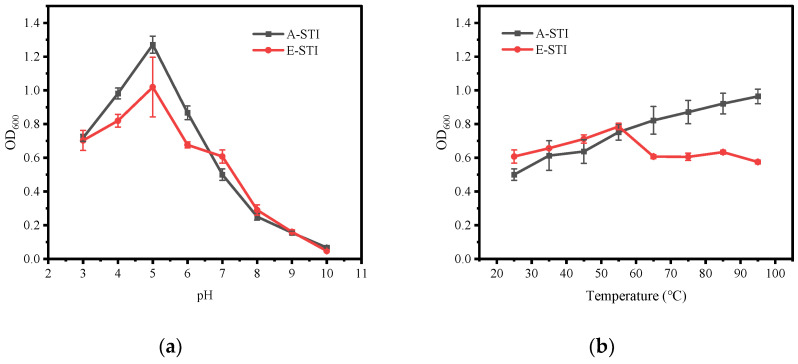
Effect of temperature and pH on the turbidity of STI: (**a**) effect of pH on the turbidity of STI; (**b**) effect of temperature on the turbidity of STI.

**Figure 6 molecules-29-02613-f006:**
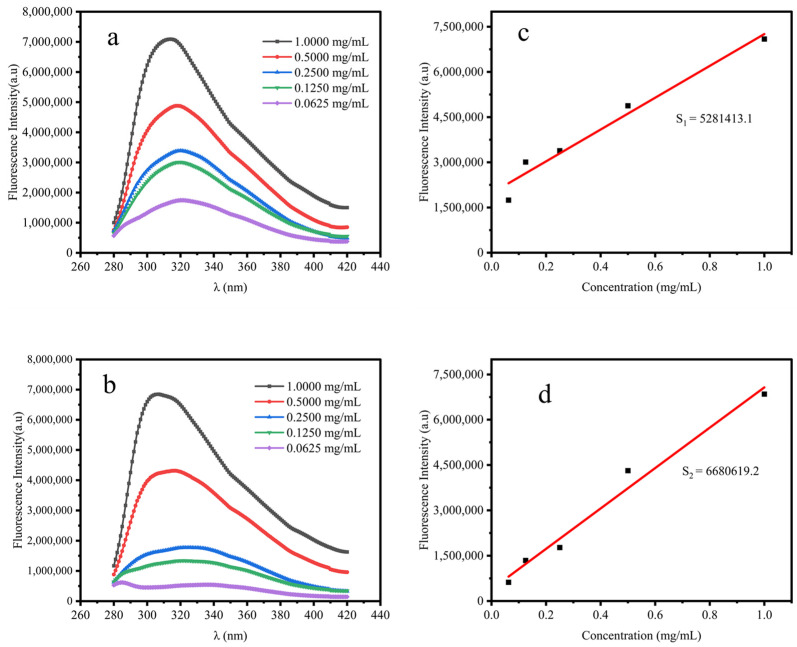
Surface hydrophobicity analysis of STI: (**a**) fluorescence intensity of A-STI; (**b**) fluorescence intensity of E-STI; (**c**) hydrophobicity index of A-STI; (**d**) hydrophobicity index of E-STI.

**Figure 7 molecules-29-02613-f007:**
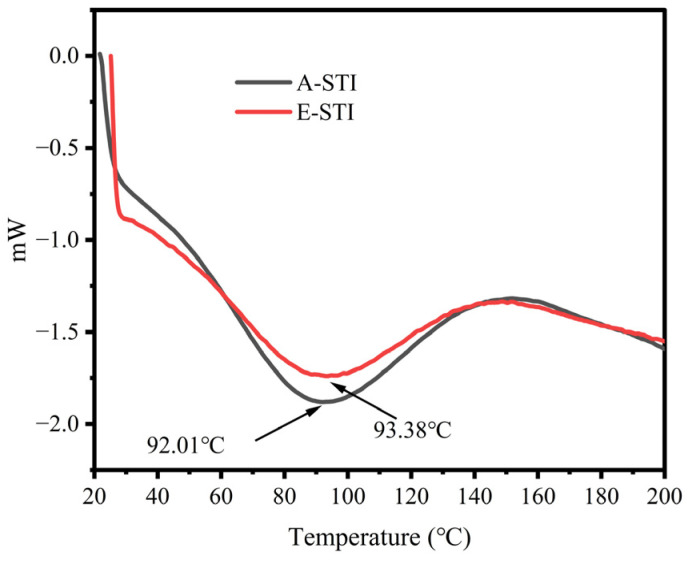
Thermal properties of A-STI and E-STI.

**Figure 8 molecules-29-02613-f008:**
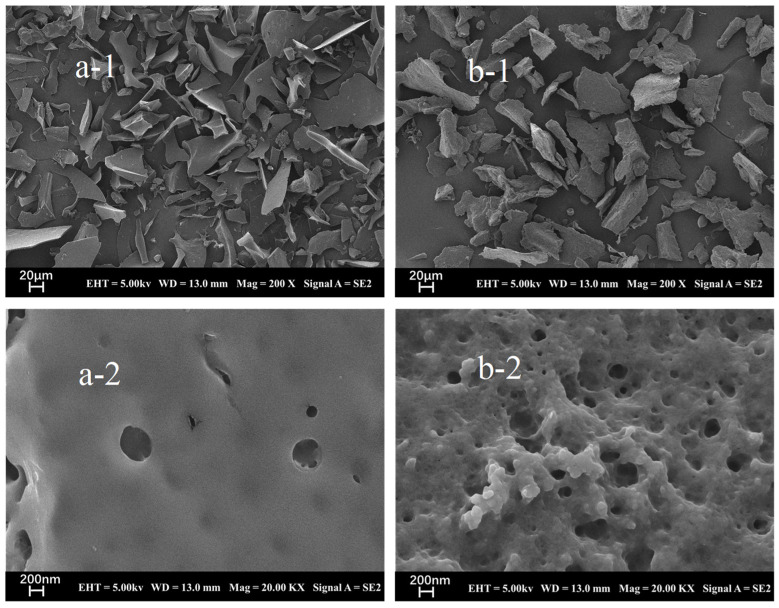
SEM micrographs of A-STI ((**a-1**): 200×, (**a-2**): 20,000×) and E-STI ((**b-1**): 200×, (**b-2**): 20,000×).

**Table 1 molecules-29-02613-t001:** The yield of STI and protein recovery rate.

Precipitation Agent	Yield (mg/mL)	Protein Content (%)	Recovery Rate (%)
Soybean whey	24.90 ± 0.15	17.19 ± 0.44	100.00
Ammonium sulfate (%)			
20	0.317 ± 0.047 ^g^	95.33 ± 0.48 ^b^	7.06 ± 0.28 ^e^
30	0.950 ± 0.041 ^f^	98.00 ± 0.55 ^a^	21.75 ± 0.69 ^d^
40	2.900 ± 0.071 ^c^	94.84 ± 0.84 ^b^	64.27 ± 0.57 ^c^
50	3.567 ± 0.131 ^b^	93.24 ± 0.37 ^b^	77.71 ± 0.11 ^b^
60	3.833 ± 0.332 ^a^	93.11 ± 0.75 ^b^	83.39 ± 0.03 ^a^
Ethanol (%)			
20	1.533 ± 0.062 ^e^	94.00 ± 2.36 ^b^	33.67 ± 0.85 ^e^
30	2.517 ± 0.085 ^d^	97.89 ± 0.65 ^a^	57.57 ± 0.44 ^d^
40	2.933 ± 0.062 ^c^	93.12 ± 0.44 ^b^	63.81 ± 0.30 ^c^
50	3.317 ± 0.062 ^b^	92.69 ± 0.77 ^b^	71.83 ± 0.64 ^b^
60	3.983 ± 0.340 ^a^	93.70 ± 0.27 ^b^	87.20 ± 0.25 ^a^

Results are presented as mean values ± standard deviation (*n* = 3). Different letters on the same column indicate significant differences (*p* < 0.05) among the different conditions.

**Table 2 molecules-29-02613-t002:** Gel filtration chromatogram peak areas for precipitated protein with different ammonium sulfate saturation (A) and ethanol content (E).

Sample	Peak 1 Area (%)	Peak 2 Area (%)	Peak 3 Area (%)
KTI	0	53.95	46.05
BBI	0	0	100.00
A-30%	18.23	57.00	24.77
A-40%	14.08	49.48	36.44
A-50%	31.84	33.04	35.13
A-60%	24.07	31.82	46.57
E-20%	27.99	43.96	28.05
E-30%	31.23	36.30	32.47
E-40%	27.20	38.30	34.50
E-50%	36.60	36.40	27.01
E-60%	21.78	29.72	49.07

## Data Availability

The data presented in this study are available on request from the corresponding author.

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
