# Peer review of "Enrichment of Trypsin Inhibitor from Soybean Whey Wastewater Using Different Precipitating Agents and Analysis of Their Properties"

_molecules, 2024, doi:10.3390/molecules29112613_

Round 1

Reviewer 1 Report

Comments and Suggestions for Authors

line 145 Does the m refer to tha mass of the precipitate?  

line 197 Define ANS solution

In Data Analysis: Was the normal behavior of the results verified? Was a mean difference analysis performed?

are there significant differences with respect to the Protein contents (%) and Recovery rate (%)?

With the values measured in DSC, is it possible to calculate the change in enthalpy?. It could be help to explain the stability

The introduction and Materials and Methods are clear

The result are very well written, detailed an clear. However, the discussion is insufficient. Not compared with similar research in all analyzes. 

How is the behavior of solubility, turbidity, yield and other properties in other soybean trypsin inhibitor (Other research)?,

How does the DSC change (Other research)?,

There is a lack of explanations for the behavior of the samples in the different tests

Reviewer 2 Report

Comments and Suggestions for Authors

This paper presents an interesting study on the enrichment of soybean trypsin inhibitor (STI) from soybean whey wastewater using ammonium sulfate salting out and ethanol precipitation methods. The authors provide a comprehensive analysis of the physicochemical properties of STI obtained through these methods. However, there are some areas that require attention to improve the clarity, rigor, and completeness of the study.

In the introduction section: there should be a more in-depth discussion on the environmental implications of soybean whey disposal and how STI recovery contributes to sustainability.

The novelty of this work should be clearly stated at the end of the introduction section.

In the materials and methods: To enhance the clarity and comprehensibility of the experimental methodology, it would be highly beneficial to include a schematic diagram that outlines the steps of the experimental work. This diagram should visually represent the workflow from the preparation of soybean whey concentrate to the final analysis of STI properties. A well-constructed diagram will help readers to better understand the sequence of procedures and the relationships between different stages of the experiment, thereby improving the reproducibility of the study

In the results and discussion section, most specifically in the thermal characterization section (Section 3.8), the significance of the thermal transition temperatures observed for A-STI and E-STI should be discussed in the context of potential industrial applications. How do these temperatures compare to other protein isolates used in food processing?

The results section is comprehensive, but the discussion could benefit from a more critical comparison with existing literature. How do the findings align or contrast with previous studies on STI recovery?

In the conclusion section, future research directions should be outlined. What are the next steps to further this research? For example, are there additional purification steps that could enhance STI quality, or are there other by-products that could be explored?

References: please add 2024 published papers to your reference list.
